# Characterizing ERA-interim and ERA5 surface wind biases using ASCAT

Maria Belmonte Rivas[1,2], Ad Stoffelen[1]

[1]Royal Netherlands Meteorology Institute (KNMI), de Bilt, 3731GA, The Netherlands
[2]Instituto de Ciencias del Mar (ICM), Consejo General de Investigaciones Cientificas (CSIC), Barcelona, 08003, Spain

*Correspondence to*: Ad Stoffelen (ad.stoffelen@knmi.nl)

**Abstract.** This paper analyses the differences between ERA-Interim and ERA5 surface winds fields relative to ASCAT ocean vector wind observations, after adjustment for the effects of atmospheric stability and density, using stress equivalent winds (U10S), and air-sea relative motion using ocean current velocities. In terms of instantaneous RMS wind speed
agreement, ERA5 winds show a 20% improvement relative to ERA interim, and a performance similar to that of currently operational ECMWF forecasts. ERA5 also performs better than ERA-interim in terms of mean and transient wind errors, wind divergence and wind stress curl biases. Yet, both ERA products show systematic errors in the partition of the wind kinetic energy into zonal and meridional, mean and transient components. ERA winds are characterized by excessive mean zonal winds (westerlies) with too weak mean poleward flows at mid-latitudes, and too weak mean meridional winds (trades)
in the tropics. ERA stress curl is too cyclonic at mid and high latitudes, with implications for Ekman upwelling estimates, and lack detail in the representation of SST gradient effects (along the equatorial cold tongues and WBC jets) and mesoscale convective airflows (along the ITCZ and the warm flanks for the WBC jets). It is conjectured that large-scale mean wind biases in ERA are related to their lack of high frequency (transient wind) variability, which should be promoting residual meridional circulations in the Ferrell and Hadley cells.

## 1 Introduction

Ocean surface wind stress and the associated heat and momentum fluxes play an important role in driving surface and deep ocean circulation. Surface wind stress modulates the amount of energy available for the ocean gyres in terms of Ekman transport and pumping [Kelly, Dickinson and Yu, 1999], ocean stirring by vertical turbulent mixing [Chen et al., 1999] and deep convection responses [Pickart et al., 2003], [Condron and Renfrew, 2013]. Many ocean models in marine forecasting
centers use ECMWF-based wind inputs for ocean forcing, including their associated biases. Some aspects of systematic error in surface winds from reanalyses have been described already, such as their defective mesoscale variability in the extra-tropics [Gille, 2005], lacking small-scale features relevant for the representation of topographic or SST gradient effects [Risien and Chelton, 2008] or generally excessive zonal winds [Chaudhuri et al., 2013]. In this paper we make another attempt to characterize the differences between observed ASCAT and ERA5/ERA-Interim surface wind fields, analyzing

zonal and meridional errors in terms of mean and transient components, and attempting to interpret those differences in terms of surface wind divergence and wind stress curl errors.

ASCAT winds have been reprocessed to obtain consistent accuracy [Verhoef et al., 2017], stability [Belmonte et al., 2017] and state-of –art algorithms [Stoffelen et al., 2017; de Kloe et al., 2017; Vogelzang et al., 2017]. Triple collocation analyses [Vogelzang et al., 2011] with moored buoys reveal that global random ASCAT wind component errors on the scatterometer measurement scale are about 0.7 m/s with negligible bias. We further note that the spatial resolution of ASCAT is about 25 km, while that of the best model product is over 100 km [Vogelzang et al., 2011], which obviously will impact spatial gradient amplitudes.

Recently, King et al. (2017) reported on the association of ASCAT wind convergence and divergence with rain events in tropical moist convection and on the fast decorrelation time (< 50 minutes) in such cases, while such associations are not found in collocated ECMWF spatial wind derivatives. Therefore, since tropical moist convection is widespread and spatial wind derivatives are relevant for ocean forcing, this is a good motivation to elaborate on the wind and stress spatial derivatives.

Accurate comparisons between collocated scatterometer and reanalysis winds require adjustments to account for the effects of atmospheric stability, density and ocean currents [Chelton and Freilich, 2005; de Kloe et al., 2017]. We use ERA-Interim and ERA5 first-guess stress-equivalent winds (U10), corrected for atmospheric stability and density, and space and time-collocated to reprocessed ASCAT-A Level-3 stress-equivalent winds over 2016. The influence of ocean currents in the observed wind differences is described in a separate section.

## 2 Methods

A common statistical metric used to assess the agreement between the wind fields described by model [$u_{nwp}$, $v_{nwp}$] and observational [$u_{scat}$, $v_{scat}$] wind components is the RMS wind vector difference, $RMS_1$, defined as:

$$RMS_1 = \sqrt{\langle (u_{scat} - u_{nwp})^2 + (v_{scat} - v_{nwp})^2 \rangle} \tag{1}$$

Although simple and compact, this metric is problematic in that it mixes zonal and meridional, mean and transient biases into a single quantity, which may not be informative when it comes to understanding the root sources of the wind field differences. Instead, one may consider that the zonal and meridional components of the surface wind at a certain location are described as:

$$u(t) = \langle u \rangle + u'(t)$$

$$v(t) = \langle v \rangle + v'(t) \tag{2}$$

Where <u> and <v> represent the time-means, e.g., annual, and u' and v' describe the variability of the wind components around the annual means, which give account of the mesoscale, synoptic scale, planetary scale and seasonal wind variability at that location. The total kinetic energy TKE of the surface wind field results from the sum of the mean kinetic energy (MKE) associated with the annual mean wind, and the eddy kinetic energy (EKE) associated with wind variability as:

$$5 \quad TKE = \frac{1}{2}\langle u^2 + v^2 \rangle = \text{MKE} + \text{EKE} , \tag{3}$$

Where

$$MKE = \frac{1}{2}(\langle u \rangle^2 + \langle v \rangle^2)$$

$$EKE = \frac{1}{2}(\langle u'^2 \rangle + \langle v'^2 \rangle) \tag{4}$$

One may consider how the total wind energy at a given location is partitioned into separate mean (steady) and transient (eddy) components, since they affect the ocean circulation and its gyres differently. Steady wind stresses are associated with large-scale upwelling/downwelling and Ekman transport in the global oceans. Transient wind stresses, which are associated with the development of surface and internal wave motions, inertial currents and transient upwelling/downwelling events, mainly contribute in a time-integral sense to vertical mixing and the development of the mixed layer. The representation of the zonal and meridional annual mean (steady) winds can be defined as:

$$u_m = \langle u \rangle$$

$$v_m = \langle v \rangle \tag{5a}$$

And the contribution of the transient (eddy) winds to the total wind kinetic energy at the surface can be expressed via the annual mean equivalent transient (eddy) wind components, defined as:

$$u_e = \sqrt{\langle u'^2 \rangle}$$

$$v_e = \sqrt{\langle v'^2 \rangle} \tag{5b}$$

Using the partition into mean and transient wind components in Eq.(5a-b), the total agreement between the wind fields can be now expressed as:

$$20 \quad RMS_2 = \sqrt{\left(u_{m,scat} - u_{m,nwp}\right)^2 + \left(u_{e,scat} - u_{e,nwp}\right)^2 + \left(v_{m,scat} - v_{m,nwp}\right)^2 + \left(v_{e,scat} - v_{e,nwp}\right)^2} \tag{6}$$

The metric $RMS_2$ based in statistical mean and transient (eddy) wind components ($u_m$, $u_e$) defined in Eq.(6) is different from the metric $RMS_1$ based in instantaneous wind field differences defined in Eq.(1) in that it is insensitive to the temporal decorrelation of the original wind fields (i.e., the time of arrival of wind perturbations does not matter, but their amplitudes

do) and less sensitive to random model and observational noise components (see Annex), which is beneficial for the study of systematic biases.

Finally, one may also compare the representation of the spatial wind field derivatives, wind divergence and wind stress curl, using partitioning into mean (steady) and transient (eddy) components, as introduced above. The annual means of the field derivatives, which are the quantities associated to the mean (steady) winds, are defined as:

$$div_m(\vec{u}) = \langle div(\vec{u}) \rangle$$

$$curl_m(\vec{\tau}) = \langle curl(\vec{\tau}) \rangle$$

and the transient (eddy) quantities defined from the departures from the annual means as:

$$div_e(\vec{u}) = \sqrt{\langle \left( div(\vec{u}) - div_m(\vec{u}) \right)^2 \rangle} \tag{7a}$$

$$curl_e(\vec{\tau}) = \sqrt{\langle \left( curl(\vec{\tau}) - curl_m(\vec{\tau}) \right)^2 \rangle} \tag{7b}$$

## 2.1 ASCAT observations

The ASCAT surface winds are downloaded from the Copernicus Marine Environment Monitoring Service (CMEMS) corresponding to the Global Ocean Daily Gridded Reprocessed Level-3 sea surface (stress-equivalent U10S) winds from the ASCAT-A scatterometer (product: WIND_GLO_WIND_L3_REP_OBSERVATIONS_012_005) for ascending orbits at 25 km resolution (dataset: KNMI-GLO-WIND-L3-REP-OBS_METOP-A_ASCAT_25_ASC) [de Kloe et al., 2017]. Only upstream L2 swath observations that have passed the KNMI Quality Control are used in the CMEMS L3 gridded product. The CMEMS L3 products also contain wind stress, curl and divergence fields. Note that ascending orbits correspond to a local solar time equator crossing (LTAN) of the sun-synchronous MetOp satellite of 21:30 in the evening. This is, the ERA diurnal cycle is only collocated and differenced around this time of day, though without any time or space sampling errors.

## 2.2 ERA surface winds

The CMEMS L3 global wind product also contains gridded model winds from ECMWF, sampled and processed in exactly the same way as the scatterometer gridded fields, and subject to identical space and time sampling errors. In this manner, ERA-Interim and ERA5 surface winds have been space and time-collocated to the ASCAT observations. Apart from a neutral wind correction for atmospheric stratification (U10N), an air mass density correction has been applied to the model winds to obtain a better correspondence to scatterometer stress-equivalent wind (U10S) measurements [de Kloe et al., 2017]. The ERA-Interim first-guess winds, featuring a spatial grid of 79 km, come from 3-hourly forecasts based on 12-hourly analyses centered at 0 and 12 UTC. The ERA5 first guess winds come from 1-hourly forecasts based on 12-hourly analyses

centered at 6 and 18 UTC, with an improved spatial grid of 31 km. The model wind vector components are quadratically interpolated in time and linearly interpolated in space to match the ASCAT satellite observations.

## 2.3 Ocean currents

With an eye on applying a correction for relative ocean motion, the ocean surface velocity fields are also downloaded from CMEMS (product: MULTIOBS_GLO_PHY_REP_015_004). This product corresponds to the Global Total Surface and 15 m Current (Copernicus-Globcurrent) from Altimetric Geostrophic Current and Modelled Ekman Current Reprocessing (dataset: uv_rep_hourly) with 3-hourly fields of zonal and meridional ocean surface velocity on a 25 km grid, interpolated linearly in space and time to match the ASCAT satellite observations. The total velocity fields are obtained by combining geostrophic surface currents derived from satellite altimetry and modelled Ekman currents at the surface and 15 m depth using ECMWF ERA-Interim wind stress [Rio et al., 2014].

## 3 Results

### 3.1 ASCAT to ERA differences

Figure 1 shows the instantaneous RMS wind speed differences, calculated following Eq.(1), between ASCAT observations and space and time-collocated first-guess U10S winds from ERA-Interim, ERA5 and the ECMWF operational forecasts over 2016. In terms of instantaneous RMS wind speed agreement to satellite observations, the performance of the ERA5 surface winds closely approaches that of the operational ECMWF forecast model, and improves the performance of the ERA-Interim product by over 20%.

Figure 2 shows the differences in the partition of mean and eddy kinetic energies in ASCAT surface winds relative to the ERA-Interim and ERA5 products. The surface winds in the reanalyses consistently appear to have too much energy in their mean flows and too little energy in their transient (eddy) activity, the problem being particularly acute over the mid-latitude westerlies. In ERA5, the mean winds have slowed down somewhat (note the different vertical scales in Fig.2), reducing the mean kinetic energy differences to ASCAT by about one-half, and eddy activity has increased (particularly in the tropics), showing a closer agreement to ASCAT observations. Yet eddy activity in ERA5 still looks defective relative to ASCAT, particularly over the mid-latitudes. In the tropics, the mean kinetic energy in the ERA surface winds appears to be correct to first order, although the partition into zonal and meridional components is biased relative to that of ASCAT, as we shall see next.

Figures 3-4 show the differences in zonally averaged zonal and meridional annual mean surface winds in ASCAT observations relative to the ERA Interim and ERA5 products over 2016, highlighting the presence of systematic mean differences of up to 0.5 m/s in the zonal and meridional components. The systematic mean differences are very stable in

time, with an inter-annual variability of about 0.1 m/s (not shown), and consistently show excessive mean model westerlies in the mid-latitudes (poleward of 30º) and excess mean model easterlies in the tropics (equatorward of 30º). The excessive mean model westerlies at mid-latitudes are connected with insufficient mean model meridional (poleward) flows between 30º and 60º in both hemispheres. In the tropics, the excessive mean model easterlies (particularly strong along the equator) are apparently connected with insufficient mean model equatorward flows (actually converging toward the annual mean ITCZ centered around 6º N, but showing a negative kink slightly south of the equator, which is related to air-sea interaction over the equatorial cold tongue, as we will see later).

A look at the geographical distribution of annual mean wind differences illustrates the location of the errors that we see in the zonally averaged differences. Figure 5 shows the global maps of annual mean wind differences between the ASCAT and ERA products, along with the annual mean wind from ASCAT to aid the interpretation. Note that the spatial error patterns in ERA5 are very similar to those found in ERA Interim, only much reduced in amplitude (close to about 50% in the zonal component, but showing less improvement in the meridional). For the zonal component (left panels in Fig.5), negative errors (blue colors) in the mid-latitudes indicate excess model westerlies, and positive errors (red colors) in the tropics indicate excess model easterlies. For the meridional component (right panels in Fig.5), negative/positive errors (blue/red colors) in the northern/southern tropics indicate defective mean equatorward winds, while more extended positive/negative errors (red/blue colors) in the northern/southern mid-latitudes indicate defective poleward flows.

The structure of zonal and meridional mean errors in the tropics is generally characterized by insufficient model meridional wind convergence (towards the annual mean ITCZ) and uniformly excessive zonal easterly model flows. This general case is modulated in the eastern tropical Pacific (and Atlantic) by air-sea interaction effects over the equatorial cold tongue [Chelton et al., 2001]. The SST-gradient effect describes how surface winds dynamically respond to SST modification and associated ocean heat flux changes [O´Neill, 2012; Skyllingstad et al., 2007]. As the dominant south-easterlies blow into the ITCZ, they are bound to decelerate when they first cross the cold SST front south of the equator, and then accelerate as they cross the warm SST front slightly north of the equator. This air-sea interaction will also produce stress curl and wind divergence features related to the cross-wind and downwind SST gradients, respectively [O'Neill et al., 2003]. The under-representation of this SST gradient effect in the ERA products explains why the typically defective model southerly winds of the eastern tropical Pacific will appear excessive south of the equator (see blue band in left panel of Fig.5) and more defective north of the equator (see red band in left panel of Fig.5), while the typically excessive model easterly wind will appear more excessive below the equator (see red band in right panel of Fig.5) and defective above the equator (see blue band in the right panel of Fig.5), all relative to the ASCAT observations.

Figure 6 shows the global maps of annual mean transient (eddy) wind differences between ASCAT and ERA5, along with the annual mean transient (eddy) zonal and meridional winds from ASCAT. Note that transient wind variability dominates outside the tropics, where warm SST fronts carried by WBCs provide sufficient energy to generate extratropical cyclones along the zonally elongated storms tracks that feed on mid-latitude westerlies. These transient motions are formed typically

in the wintertime off the eastern continental seaboards in the Atlantic, Pacific and Indian oceans, and all year round around the ACC, which are typical baroclinic growth regions associated with large meridional SST gradients [Sampe and Xie, 2007] [Booth et al, 2017]. We note that model wind variability is overall defective in the zonal and meridional components at mid-latitudes (red colors in bottom panel of Fig.6), with locally enhanced biases (defect) along the WBCs (i.e., Agulhas Return Current, Brazil Current, Gulf Stream and Kuroshio Extension) and the Antarctic Circumpolar Current (ACC), and particularly biased (defect) meridional variability along the ITCZ in the tropics (coinciding with the maximum mean wind divergence). We also note that the wind variability in ERA5 is enhanced relative to that in ERA Interim, although still less intense overall than in the observations. The inability of reanalyses to reproduce higher frequency (mesoscale and synoptic scale) wind variability implies underestimation of atmospheric forcing at the air-sea boundary, particularly along the extra-tropical storm tracks, with detrimental consequences for ocean forcing [Condron et al., 2008] [Laffineur et al., 2014] and the representation of air-sea interaction in coupled models.

Figure 7 shows the global maps of annual mean and transient (eddy) stress curl from ASCAT, along with the differences to ERA-Interim and ERA5 in 2016. We first note that the mean stress curl in the ERA products is more cyclonic (blue colors in NH, red colors in SH) at mid and high latitudes (poleward of $30^{o}$) than in observations. Ekman upwelling is related to the mean stress curl of the surface wind via curl($\tau/\rho_0 f$), where $f$ is the Coriolis parameter and $\rho_0$ is a reference ocean density. The fact that model mean stress curl in ERA winds is more cyclonic than in observations implies that model Ekman upwelling will be overestimated at high latitudes (in the subpolar gyres), and downwelling will be underestimated in the mid-latitudes (in the subtropical gyres) relative to observations. At the same time, the eddy stress curl in the ERA products is less intense than in observations, which is also suggestive of missing mesoscale turbulence in the reanalysis winds. By conservation of momentum, we know that tropical air masses must acquire (relative) anticyclonic vorticity as they move poleward [Holton, 2004]. In the reanalysis products, surface air masses remain too cyclonic at mid to high latitudes, which may be related to defective poleward transport and diffusion of anticyclonic momentum by mesoscale turbulence. Over the cold tongues in the eastern tropical Pacific and Atlantic oceans, the SST gradient effect that we identified in the mean wind differences also leaves a signature in the mean stress curl differences. The pattern of stress curl in the tropics is dominated by a band of positive curl along 5N-10N where the northeast trades build to the north of the ITCZ, and a narrow strip of positive curl just north of the Equator sustained by the lateral gradient of wind stress generated by the acceleration of surface winds over the northern front of the cold tongue (see top left panel in Fig.7) [Chelton et al, 2001] accompanied by a more extended band of negative curl to the south. The map of mean stress curl differences indicate that the signature of wind curl associated with the equatorial cold tongue is underrepresented in the ERA products, with model defective positive curl in the northern front (reddish colors) and model defective negative curl in the southern front (bluish colors, see bottom left panels in Fig.7). A more detailed (zoomed) depiction of the underrepresentation of model stress curl over the cold tongue in the eastern Tropical Pacific is shown later (see Fig 16). Finally, note that the spatial distribution of missing model wind variability in the eddy stress curl maps is very similar to that provided by the transient (eddy) winds maps everywhere except along the ITCZ -

where the signature of wind variability appears to be more associated to eddy divergence effects from unresolved airflows in rainy conditions [Lin et al., 2015], as we shall see next.

Figure 8 shows the global maps of annual mean and eddy wind divergence from ASCAT, along with the differences to the ERA-Interim and ERA5 products in 2016. The pattern of differences in mean wind divergence is subtle, but shows defective model mean wind divergence (red colors in Fig.8) over the subtropical gyres in the mid-latitudes, and slightly defective model mean wind convergence (blue colors in Fig.8) over the subpolar gyres at high latitudes. This dipolar pattern of defective model mean wind divergence/convergence could be related to missing model subsidence in the subtropics, and missing model uplift in the subpolar areas, which when connected to the meridional mean wind biases of the mid-latitudes, remains suggestive of a residual meridional circulation in the Ferrell cell driven by mesoscale turbulence that is underrepresented in the reanalysis winds (see Discussion).

In the tropics, the maps of ASCAT to ERA differences show defective model convergence (blue color bands in Fig.8) along the ITCZ (characterized by maximum wind convergence, and flanked by bands of positive and negative stress curl on the north and south sides) in direct connection with the meridional mean wind biases already identified. Superimposed on this, we note a signature of defective model wind divergence (red color strip in Fig.8) over the northern SST front of the cold tongue in the eastern tropical Pacific, and defective model wind convergence (blue color band in Fig.8) over the southern SST front, also identified as indicative of problems with the representation of SST gradient effects. Recall that the downwind deceleration/acceleration of southeast trades over the cold tongue is bound to result in convergence (and negative stress curl via the lateral gradient of wind stress) when crossing the southern SST front, and to form a strip of divergence (with a thin strip of positive curl) when crossing over the northern SST front [Chelton et al., 2001].

The maps of differences in eddy wind divergence (right panels in Fig.8) show defective model eddy divergence along the ITCZ, in direct connection with the band of defective meridional eddy winds (right panels in Fig.6), and along the warm flanks of the WBCs. In the tropics, deep moist convection causes wind downbursts, bringing dry air from upper levels to the ocean surface, however, strongly underestimated in ERA [King et al., 2017]. Besides continuous tropical deficits in momentum exchange, this fast process probably causes mean effects on precipitation, evaporation and ocean-atmosphere heat exchange [Chiang and Sobel, 2002]. Extreme surface wind divergence events have also been associated to downdrafts from stratiform mesoscale convective systems that organize around convective towers in the tropics [Kilpatrick and Xie, 2015], and remain indicative of problems with unresolved airflows linked to moist convection in model winds.

### 3.2 Effect of spatial resolution

The amount of energy captured in the eddy maps of Fig. 6 obviously depends on spatial resolution, which for the ASCAT data is approximately 25 km.  Recall that the ERA effective spatial resolution is less than 100 km [Vogelzang et al., 2011].

All the ERA products have been carefully collocated in space and time to match ASCAT observations, but the differences in effective spatial resolution remain problematic, not so much for the mean wind or mean derivative fields, but for the eddy fields. To further investigate this effect a spatial smoothing, using a 1.5 degree spatial filter before calculating the mean and eddy quantities, was employed. It appeared (not shown) that indeed mean (wind, curl, divergence) differences are not appreciably dependent on resolution, including transient (eddy) wind differences. On the other hand and as anticipated, transient (eddy) curl and divergence are affected by the spatial filter: both model and satellite eddy fields change, but the reduction is largest in the satellite fields.

In summary, if we force the spatial resolution of both model and satellite data to be similar (down to 100-200 km), the differences in mean wind/curl/divergence do not change. That is, meridional winds are still weak in the model, tropical convergence is still low in the model and stress curl at high latitudes is still high in the model.

## 3.3 Effect of ocean currents

Satellite scatterometers measure wind stress relative to the ocean surface velocity, making scatterometer observations the most appropriate quantity to describe ocean forcing and air-sea fluxes at the air-ocean interface [Kelly et al., 2001]. For instance, simulations show that the wind power input into the ocean is reduced by 20-35% after considering the effect of ocean currents on wind stress [Duhaut and Straub, 2006]. In this section we complete the analysis of the differences between reanalysis and scatterometer winds by considering the effect of ocean currents on the anomalies observed. Figure 9 shows the annual mean and transient (eddy) ocean surface zonal and meridional velocities according to the CMEMS ocean current (MULTIOBS_GLO_PHY_REP_015_004) product, which includes geostrophic surface currents derived from satellite altimetry and modelled Ekman currents at the surface derived using ECMWF ERA-Interim wind stress [Rio et al., 2014]. In this figure, one identifies the mean zonal component of the eastward flowing WBCs (Gulf Stream and Kuroshio Extension, along with the Agulhas Return and the Brazil-Malvinas Confluence merging into the ACC). In the tropics, one may also identify the alternating system of westward flowing North Equatorial and South Equatorial Currents (NEC and SEC, in blue colors) with the opposing North Equatorial Counter Current (NECC, in red color) and the subtle imprint of the South Equatorial Undercurrent (SEUC) slightly south of Equator, dividing the SEC into its northern SEC(N) and southern SEC(S) branches. In the mean meridional components, one can identify the eastern boundary currents (California, Peru, Benguela, Canary and Western Australia coastal currents) together with the strong signature of tropical upwelling along the Equator. We note that, aside from the seasonal and intra-seasonal variability of the equatorial current systems, most of the transient activity in ocean surface velocities can be associated with the WBCs in the extra-tropics, illustrating the impact of ocean currents on maintaining the higher frequency wind variability through their role in maintaining sharp SST fronts.

Figures 10 and 11 show the zonal and meridional mean wind differences between the reanalysis and scatterometer products obtained after applying the CMEMS correction for ocean surface velocity. The CMEMS ocean current correction is derived

from the original 3 hourly fields of zonal and meridional ocean surface velocities at 25 km resolution, which are linearly interpolated in space and time to match ASCAT observations. We find that the ocean current correction has barely no effect on meridional wind biases, but notably relieves the zonal wind biases, particularly in the SH mid-latitudes with up to 50% error reduction. The ocean current correction is altering the pattern of zonal mean wind biases in the trade regions, revealing differences that appear more realistically associated with: i) the representation of airflows connected to unresolved mesoscale convection over the ITCZ from $4^{o}$N to $10^{o}$N (with excess model easterlies in red color in the left panel of Fig.12, and defective meridional winds); and ii) the representation of SST gradient effects over the equatorial cold tongues (with excess model easterlies in red color in the left panel of Fig.12, and excess model southerlies in blue color in the right panel as the wind crosses the cold SST front south of the Equator, and defective model easterlies in blue color in left panel of Fig.12, with defective model southerlies in red color in the right panel as the flow crosses the warm SST front slightly north of Equator from 0 to $4^{o}$N) as further detailed below.

Figures 13-14 show the balance between observed and model mean and eddy kinetic energies before and after the ocean current correction (cf. Fig.2). The alleviation of zonal mean wind errors reduces the MKE differences in the mid-latitudes by about one half, but increases the MKE differences in the tropics, indicating that the model mean wind speeds in the trade regions have become weak relative to observations after the ocean current correction, which is to be mainly attributed to too weak model meridional inflows into the ITCZ (see right panel in Fig.12). We observe that EKE differences increase globally, particularly in the extra-tropics. In the Southern Ocean, the increase in EKE differences is accompanied by the largest decrease in MKE differences. After the ocean current correction, the EKE in the model relative winds, which was expected to increase due to the uncorrelated variability of geostrophic currents, becomes effectively diminished by removing the coherent (wind-driven) signature of Ekman currents. This correction is subtracting energy from the already defective model wind variability, and increasing the gap between the observed and modelled eddy kinetic energies. It is interesting to note that the patterns of MKE and EKE differences in the tropics also look more anti-symmetric after the ocean current correction, suggesting that the relationship between mean and transient model wind errors over the ITCZ has a nature similar to that found in the mid-latitudes, where missing model variability appears connected with stronger zonal flows and associated with defective meridional winds.

Finally, we look into the effects of the ocean current correction in the wind derivative fields. The magnitude of ocean divergence is negligible compared to the observed anomalies in surface wind divergence. The effect of ocean vorticity in the wind curl differences between ASCAT and ERA5 is shown in Fig.15. It does not modify the general picture of excess model cyclonicity already described at mid and high latitudes, but it does introduce quite an amount of new structure across the WBC jets and the equatorial cold tongues, in a manner such that the wind or wind curl differences align more realistically with the underlying SST fronts that sustain the ocean currents. The introduction of the ocean current correction seems to provide a more realistic portrayal of the problems with the representation of air-sea interaction effects in reanalysis winds, revealing consistently stronger (weaker) wind speeds over warm (cold) SST fronts in the observation to model differences.

Figures 16 and 17 show details of the effects of the ocean current correction in mean wind speed and stress curl differences in the eastern tropical Pacific and the Gulf Stream. In the eastern tropical Pacific, we already saw that model winds were under-representing SST gradient effects, being too weak (strong) over cold (warm) SST fronts relative to ASCAT observations. Figure 16 shows that the ocean current correction accentuates these differences, and aligns them better with the actual SST fronts that underlie the branched SEC(N) and SEC(S) currents just north and south of the Equator, enhancing the narrow strip of positive curl just north of the Equator that previous research [Kessler, Johnson and Moore, 2003] has underlined as important for the representation of equatorial ocean circulation and the actual maintenance of the SEC(N). Figure 17 shows that only after the ocean current correction is introduced in the Gulf Stream, we get to recover the expected signature of an under-represented SST gradient effect in the wind speed and wind curl differences, as observed in the eastern tropical Pacific, with coincident high SST and observed wind speeds over the Gulf Stream jet, and positive (negative) observed curl anomalies upwind (downwind) of the warm SST tongue.

## 4 Discussion

The most outstanding large-scale feature in the observed-to-model wind field differences at mid-latitudes is the insufficient transient wind variability in the ERA products, which can be reasonably attributed to its lower spatial resolution. The lower model transient wind activity can be associated with: a) excessive model zonal mean winds, b) defective poleward flows, c) excess cyclonic stress curl and d) defective subtropical (subpolar) divergence (convergence). In order to connect all these features together, one may reasonably postulate the idea that some additional transient wind variability (see Fig.18) should induce a residual meridional (poleward, wave driven) circulation in the Ferrel Cell, which would a) subtract energy from the mean zonal wind, b) correct for the meridional mean wind biases, c) transport anticyclonic momentum northwards to correct for the stress curl bias, and d) set off a closed circulation with subsidence at mid-latitudes (to correct for the divergence bias) and lift at high latitudes (to correct for the convergence bias).

The problem with deficient transient wind variability in the ERA products may be also connected with a long-standing problem of numerical prediction models, that of underestimation in wind turning across the boundary layer, whereby surface winds stay more aligned to the geostrophic balance above than to the pressure gradient below, somehow implying that model winds carry enhanced zonal components and reduced meridional flows. [Sandu et al., 2013] report that turbulent diffusion (already enlarged to account for sub-grid mesoscale variability and other factors in the model wind fields) is already too large and detrimental to the representation of stable boundary layers (their depths being overestimated), although it helps improve the representation of synoptic cyclones (by increasing the cross-isobaric inflow) at the expense of reducing the ageostrophic wind turning angle. Reducing turbulent diffusion would thus improve the representation of stable (boundary) layers, but be detrimental to the medium-range forecasting of cyclones. The problem is how to reconcile the fact that transient wind variability is low in the model, with the idea that turbulent diffusion is already too large. Perhaps it is not appropriate to use the (Monin-Obukhov) turbulent diffusion scheme, which accounts for turbulence generated by vertical

shear under different (thermodynamical) stability conditions, to compensate for missing mesoscale variability. The Monin-Obukhov parameterizations are observed to work well near the surface, but additional processes might be needed at higher levels in the boundary layer. Perhaps a different mechanism for surface drag, such as dynamical (baroclinic) instability, is necessary to provide stronger horizontal diffusion at low levels. A particular instance of dynamical instability occurs in (hurricane) boundary layer flows, where observations convincingly demonstrate that a large fraction of turbulent flow is organized into intense horizontal roll vortices [Foster, 2005]. The standard downgradient turbulent parameterizations cannot represent the rolls' inherently non-local and non-gradient contributions to the turbulent fluxes, because roll transport is due to an embedded secondary circulation that organizes smaller-scale turbulent eddies and advectively transports momentum, heat and moisture across the boundary layer (as opposed to downgradient diffusion). The strong sensitivity of numerical model simulations of hurricanes to boundary layer parameterizations is discussed in [Braun and Tao, 2000].

Focusing on the low latitudes, [Simpson et al., 2018] present a number of pieces of evidence that support the hypothesis of similarly missing drag on low level zonal flows. They report that in ERA-interim, analysis increments act to weaken the zonal surface winds, which is indicative of missing drag at low level, while increasing meridional convergence at low latitudes, strengthening the Hadley circulation. The missing drag mechanism in this case is conjectured to be related to the resolution of mesoscale convective airflows. Similarly to the mid-latitude case, where we conjecture the necessity of an additional turbulent drag mechanism (in the form of transient eddy curls) to erode the dynamical law of momentum conservation that restrains poleward motion, in the tropics we conjecture the need for an additional turbulent drag mechanism (in the form of transient convective airflows) to erode the thermal law of conservation of entropy that restrains vertical motion.

In all cases, we note that the model-to-satellite wind differences are limited to ascending ASCAT measurements, which correspond to nighttime (approximately 9.30 pm) conditions. The diurnal variability of surface winds certainly limits the representativity of the nighttime differences that we observe, since the ERA diurnal cycle may not be perfect. Nevertheless, the main conclusions are not expected to change for daytime conditions. Actually, the boundary layer destabilization that generally takes place during daytime is expected to increase the amount of higher frequency wind variability, which is underrepresented in ERA, and thus enhance the magnitude of the model-to-satellite differences reported here.

## 5 Conclusions

We have performed a comparison of the surface wind fields represented in the ERA-Interim and ERA5 reanalyses with ASCAT observations, and inquired into the nature of those differences in terms of zonal and meridional components of the annual mean and transient winds, wind divergence and stress curl. Before the correction for ocean surface velocity, we find that in terms of instantaneous RMS wind speed agreement to ASCAT observations, ERA5 winds show a 20% improvement relative to ERA interim, and a performance similar to that of currently operational ECMWF forecasts. Yet ERA wind fields

show systematic error patterns regarding the partition of the wind kinetic energy into zonal and meridional, mean and eddy components. More specifically:

- In terms of mean wind errors, the ERA products show excess mean zonal winds (too westerly in mid-latitudes, and too easterly in the tropics) and defective mean meridional winds (not poleward enough in mid-latitudes, not equatorward enough in the trade regions) with systematic differences of up to 0.5 m/s in the mean zonal and meridional components.

- In terms of transient wind errors, the ERA products show deficient zonal and meridional model wind variabilities, mainly over the storm tracks associated to WBCs systems and the ACC in the mid-latitudes, and missing meridional variability associated to ITCZ in the tropics.

- In terms of errors in stress curl, the ERA products are more cyclonic than observations at mid and high latitudes, with implications for Ekman upwelling estimates, that is, excess model upwelling at high latitudes, and defective Ekman downwelling at mid-latitudes. There is a spatial correlation between defective transient wind activity in the extra-tropics and defective model eddy stress curl.

- In terms of errors in wind divergence, the ERA products show deficient wind divergence in the subtropical gyres, and deficient wind convergence in the subpolar gyres, with possible implications for atmospheric vertical motion, that is, defective air subsidence at mid-latitudes and defective air lift at high latitudes. In the tropics, the ERA wind fields show defective mean wind convergence along the ITCZ, with signs of underrepresented SST gradient effects along the equatorial cold tongues (also visible in the mean stress curl differences). The ERA products are overall low in eddy wind divergence, which remains indicative of problems with unresolved airflows under moist convection conditions, particularly along the ITCZ and the warm flanks of the WBCs, interestingly correlated with defective meridional transient wind activity.

By all accounts, ERA5 performs better than ERA interim (i.e., instantaneous RMS agreement, mean and transient wind errors, stress curl and wind divergence errors). On the other hand, the remaining combination of excess model cyclonicity, insufficient eddy curl (or eddy wind) activity, and too weak mean meridional winds at the mid-latitudes may be interpreted as a sign of insufficient poleward transport and diffusion of anticyclonic momentum in the model, which when connected with the signature of too weak divergence/convergence at mid/high latitudes, remains suggestive of the need for a stronger Ferrell cell in the ECMWF model driven by mesoscale turbulence. Also, the combination of insufficient model mean wind convergence and eddy wind divergence along the ITCZ is suggestive of misrepresentation of mesoscale convection in the tropics. This is in line with anomalies observed in the ECMWF vertical wind shear climate profile, as compared to collocated high vertical resolution radiosondes (Houchi et al., 2010).

Ocean currents make a notable contribution to the mean differences between scatterometer and reanalysis winds. The ocean current correction relieves the zonal mean wind biases in the mid-latitude westerlies by up to 50%, and in the trade regions,

but with almost no effect on meridional mean wind biases. After the ocean correction, model mean winds remain stronger than observations in the mid-latitudes, but appear weaker than observations in the tropics (mostly due to the missing meridional component). The correction for ocean current is introducing new error patterns in the tropics that are more consistent with the misrepresentation of SST gradient effects over the equatorial cold tongues, and the resolution of mesoscale convection effects over the ITCZ. On the other hand, the ocean current correction is increasing the transient wind biases by subtraction of eddy kinetic energy from model wind variability via the modeled Ekman response. The signature of ocean current divergence is negligible, but the signature of ocean current curl is notable, and acts to enhance the mean stress curl differences between observations and model wind fields, particularly there where strong currents are sustained by strong SST fronts, and larger air-sea interaction effects are expected. In other words, a correction for ocean surface velocity is essential for the characterization of problems with the representation of air-sea interaction effects in ERA winds.

**Annex**

If we express the original wind components in the reanalysis $[u_{nwp}, v_{nwp}]$ and observations $[u_{scat}, v_{scat}]$ as:

$$u(t) = \langle u \rangle + u'(t) + \varepsilon(t)$$

Where we only describe the zonal component for simplicity, with $<u>$ the annual mean, $u'$ the wind variability around the annual mean, and $\varepsilon$ a measure of the random system noise. Then the $RMS_1$ metric based on instantaneous wind field differences and defined in Eq.(1) can be rewritten as:

$$RMS_1^2 = \langle (u_{scat} - u_{nwp})^2 \rangle = (\langle u_{scat} \rangle - \langle u_{nwp} \rangle)^2 + \langle u_{scat}'^2 \rangle + \langle u_{nwp}'^2 \rangle - 2\langle u_{scat}' \cdot u_{nwp}' \rangle + \langle \varepsilon_{scat}^2 \rangle + \langle \varepsilon_{nwp}^2 \rangle$$

Similarly, the $RMS_2$ metric based on statistical mean and transient (eddy) wind components and defined in Eq.(6) can be rewritten in terms of annual mean, wind variability and (to first order) random noise components as:

$$RMS_2^2 = \left( u_{m,scat} - u_{m,nwp} \right)^2 + \left( u_{e,scat} - u_{e,nwp} \right)^2$$

$$= \left( \langle u_{scat} \rangle - \langle u_{nwp} \rangle \right)^2 + \langle u_{scat}'^2 \rangle + \langle u_{nwp}'^2 \rangle - 2\sqrt{\langle u_{scat}'^2 \rangle \cdot \langle u_{nwp}'^2 \rangle} + \langle \varepsilon_{scat}^2 \rangle (1 - \sqrt{\frac{\langle u_{nwp}'^2 \rangle}{\langle u_{scat}'^2 \rangle}})$$

$$+ \langle \varepsilon_{nwp}^2 \rangle (1 - \sqrt{\frac{\langle u_{scat}'^2 \rangle}{\langle u_{nwp}'^2 \rangle}})$$

The direct comparison of the two expressions above reveals that the only differences arise from 1) the cross-correlation of the wind variability terms and 2) the sensitivity to random noise. Note that the $RMS_2$ metric does not care about the simultaneity of wind perturbations in the reanalysis and observational systems, but just about the energy that they carry. Also note that as long as the representation of the energy carried by the wind variability terms is comparable in both systems (reanalysis and observations), then there will be an effective cancellation of the system random noise components in $RMS_2$.

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

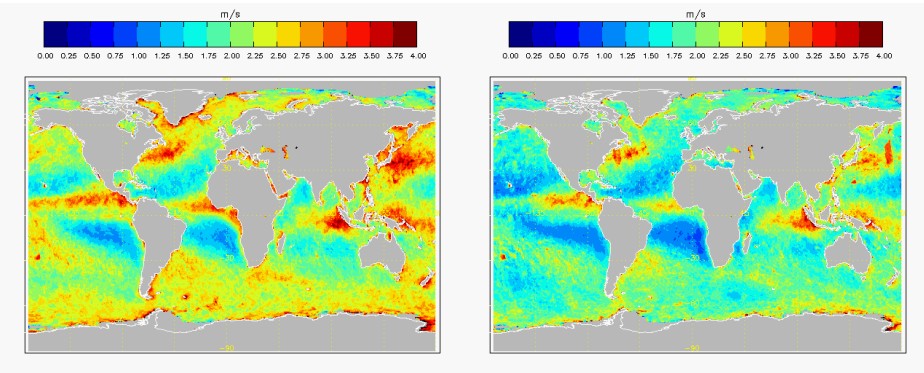

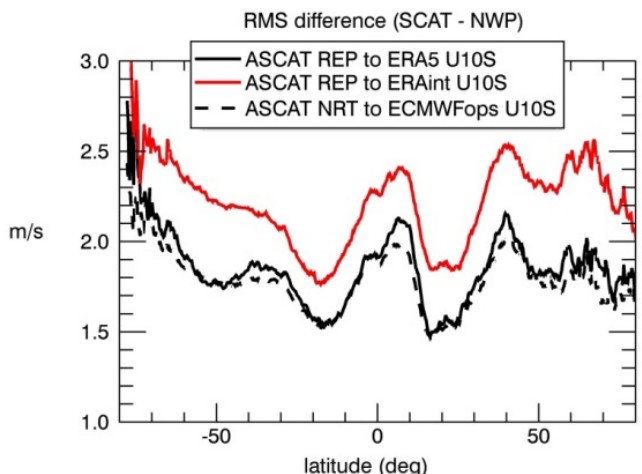

**Figure 1: Instantaneous RMS wind speed differences between ASCAT observations and ERA Interim (top left panel) or ERA5 (top right panel). The bottom panel shows the zonally averaged instantaneous RMS curves (ERA-Interim in red continuous line, ERA5 in black continuous line; and ECMWF operational forecasts in dashed black lines) first-guess winds over 2016.**

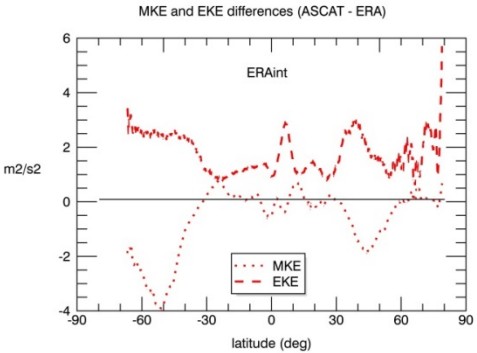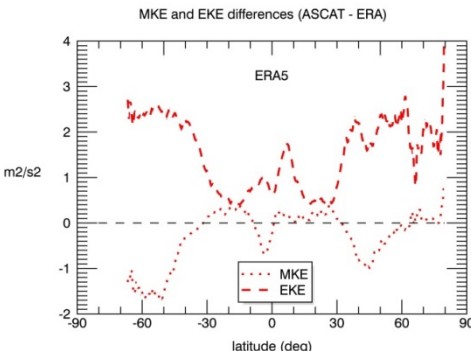

**Figure 2: Zonally averaged differences in mean kinetic energy (MKE, dots) and eddy kinetic energy (EKE, dashes) between ASCAT observations and ERA-Interim (left plot) / ERA5 (right plot) first-guess winds over 2016.**

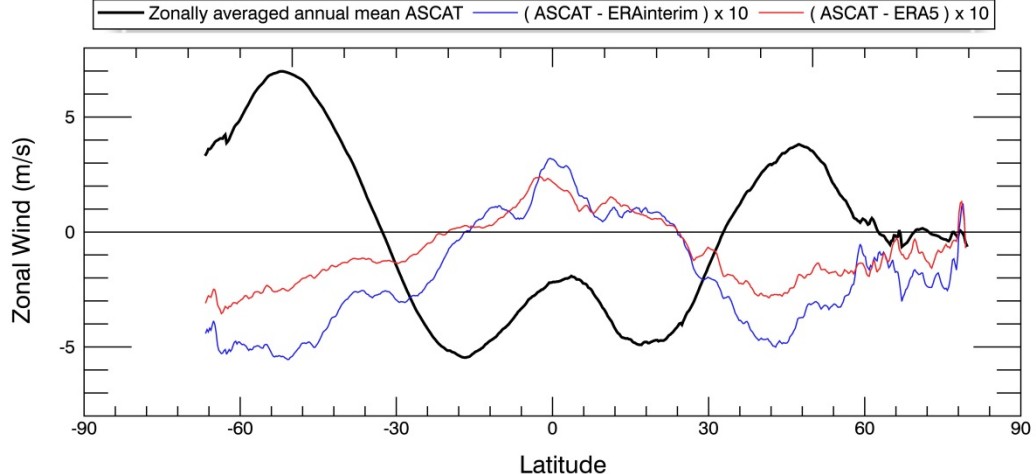

**Figure 3: Zonally averaged annual mean zonal wind from ASCAT (thick black line) and differences to ERA-Interim (blue line) and ERA5 (red line) first-guess winds over 2016. Note that differences have been scaled by 10.**

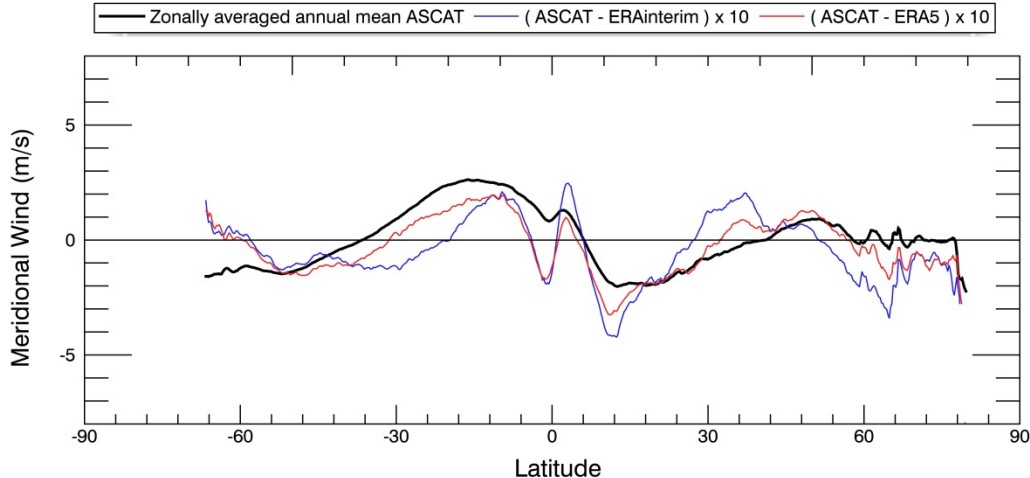

5   **Figure 4: Zonally averaged annual mean meridional wind from ASCAT (thick black line) and differences to ERA-Interim (blue line) and ERA5 (red line) first-guess winds over 2016. Note that differences have been scaled by 10.**

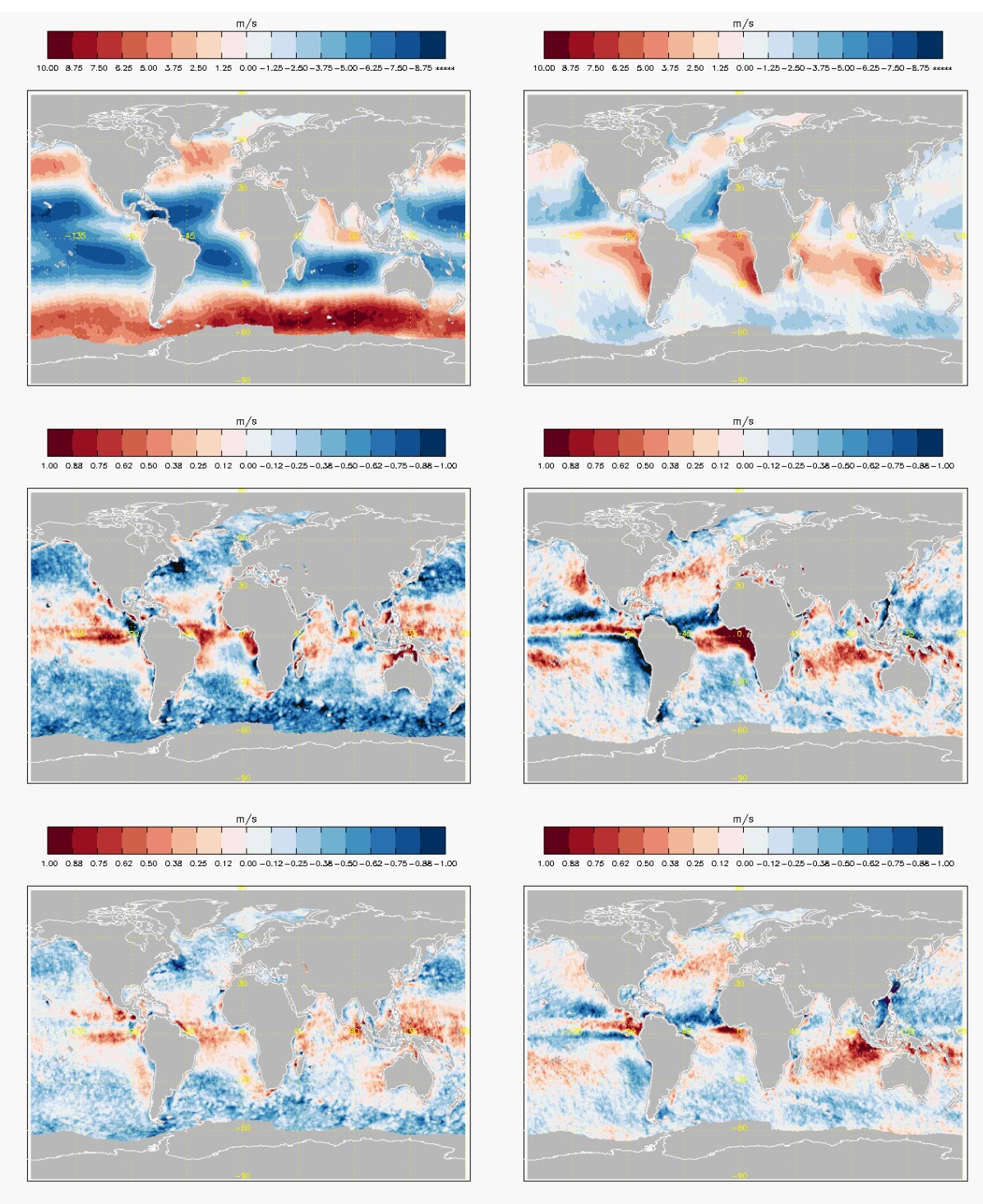

**Figure 5: Global maps of annual mean zonal (top left) and meridional (top right) wind from ASCAT and differences from ERA Interim (middle row) and ERA5 (bottom row) first-guess winds over 2016.**

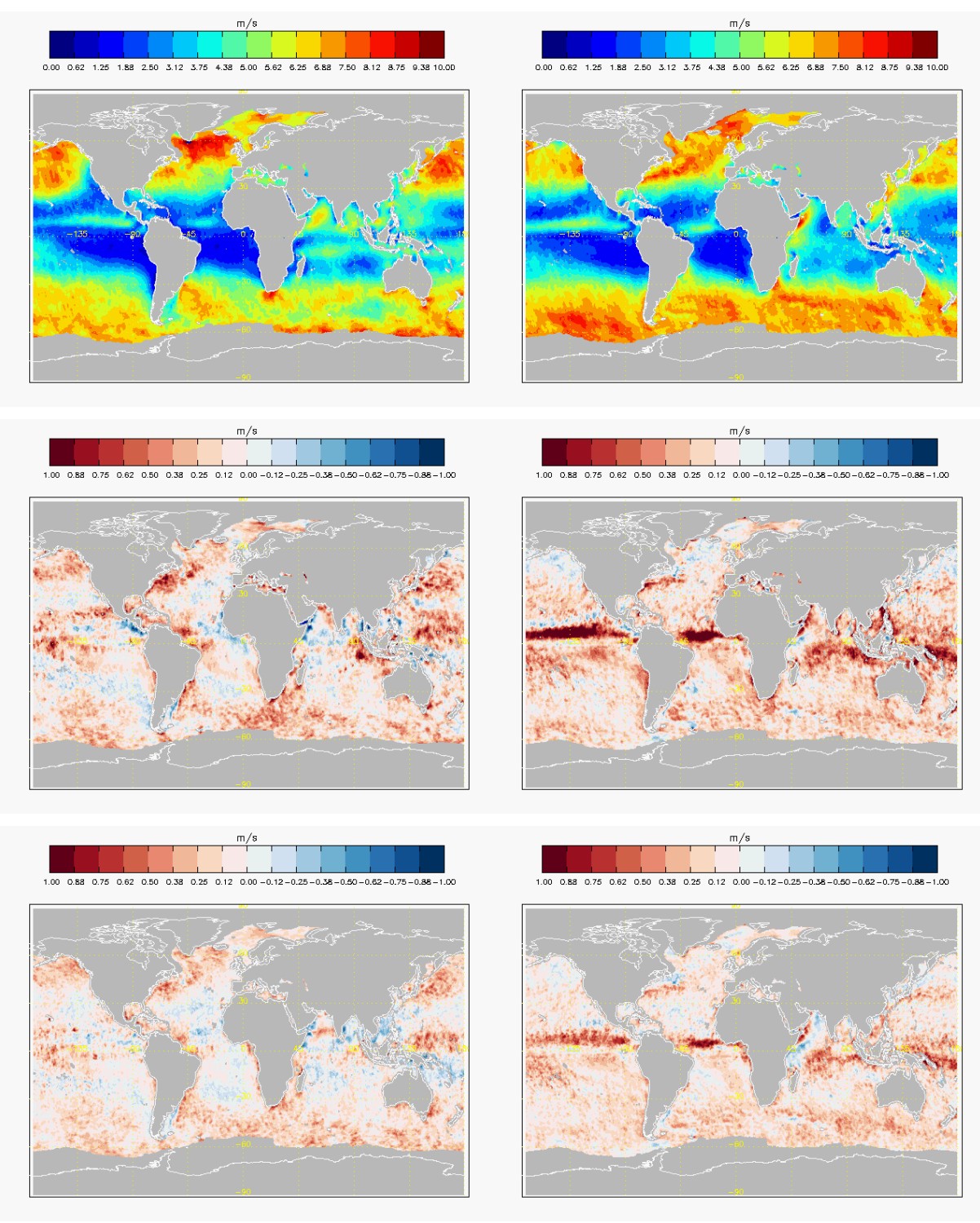

**Figure 6: Global maps of annual mean zonal (top left) and meridional (top right) transient wind from ASCAT and differences from ERA Interim (middle row) and ERA5 (bottom row) first-guess winds over 2016.**

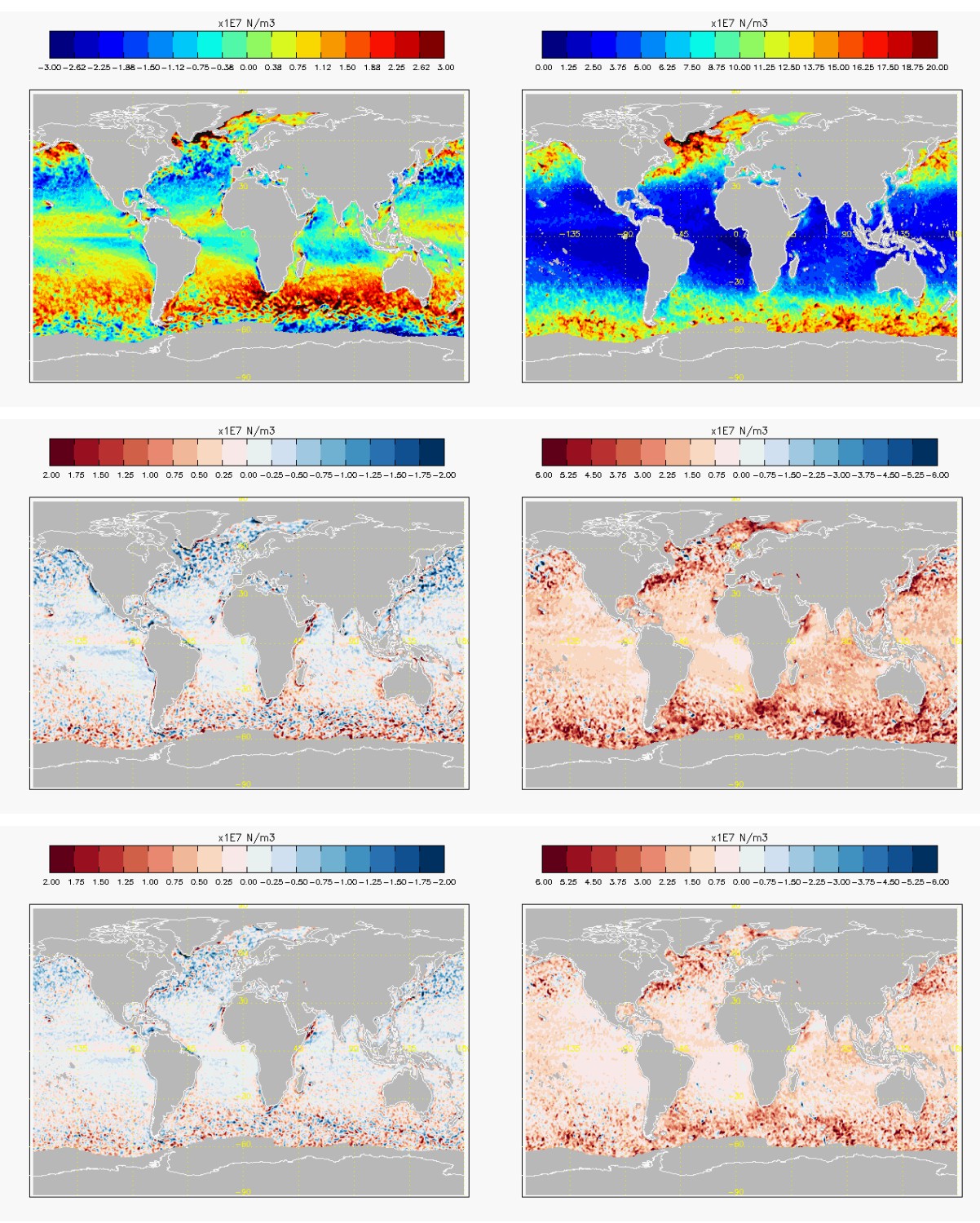

**Figure 7: Global maps of annual mean (top left) and eddy (top right) stress curl from ASCAT and differences to ERA Interim (middle row) and ERA5 (bottom row) first-guess winds over 2016. Note the color scale differences.**

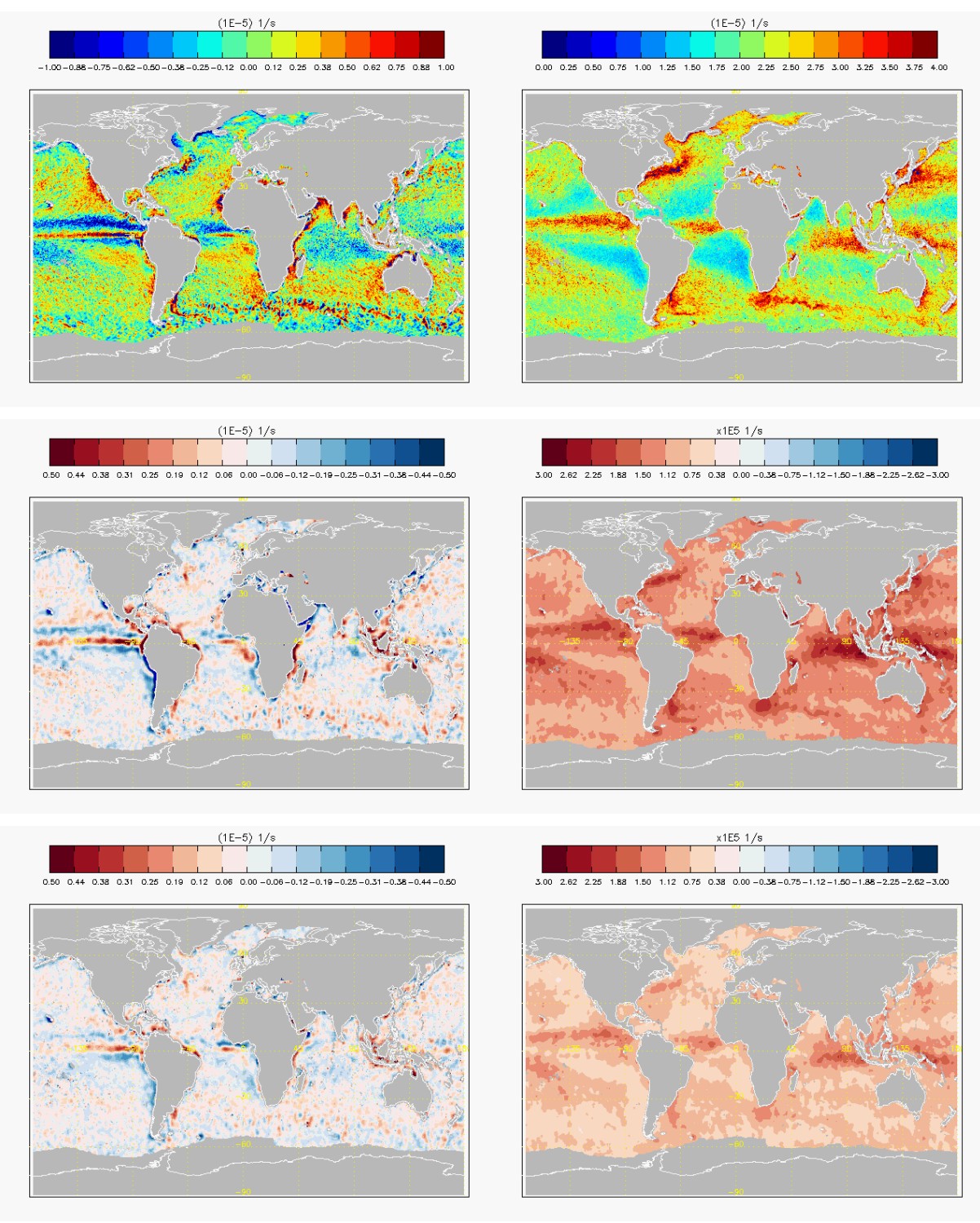

**Figure 8: Global maps of annual mean (top left) and eddy (top right) wind divergence from ASCAT and differences to ERA Interim (middle row) and ERA5 (bottom row) first-guess winds over 2016. Note the color scale differences.**

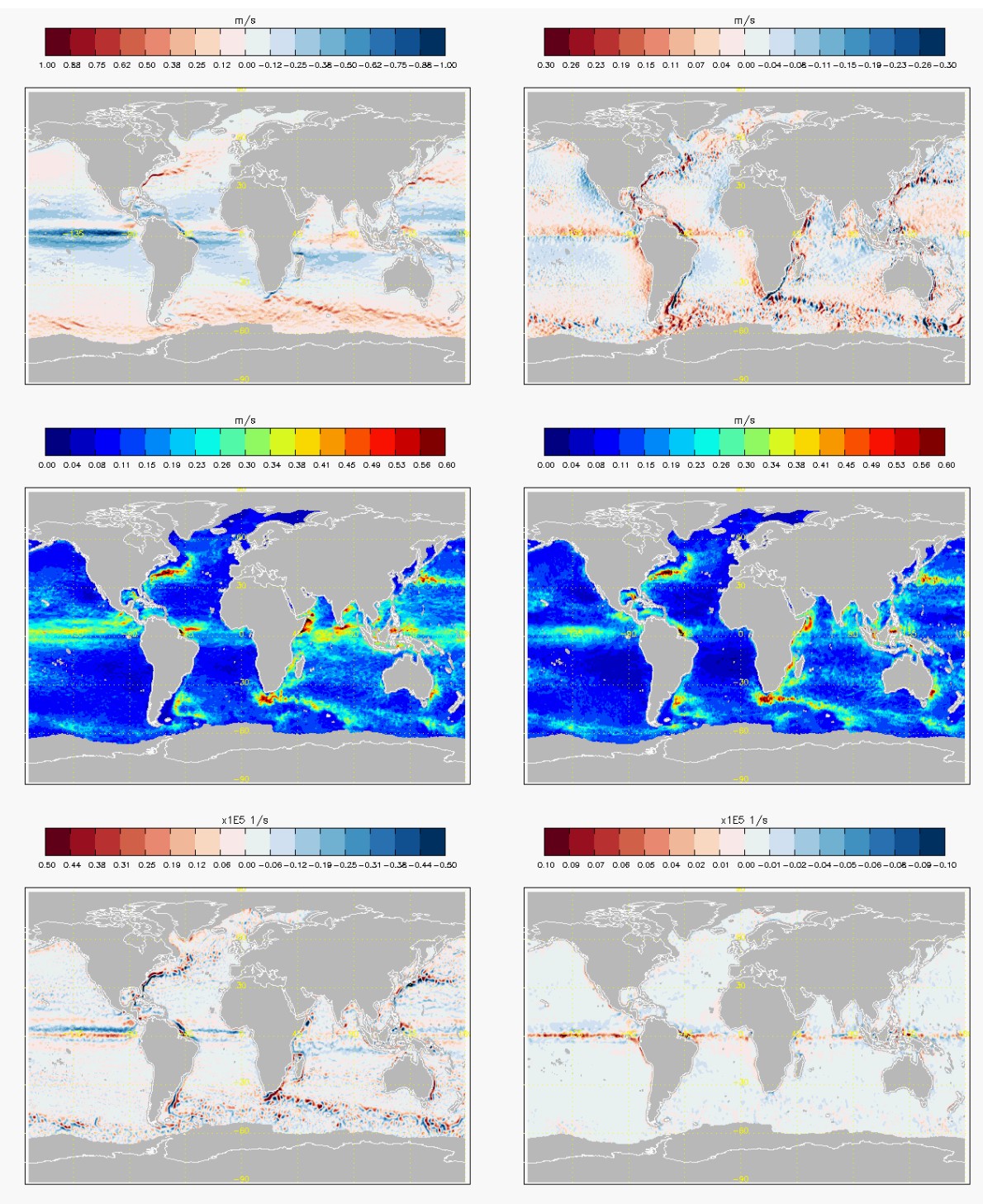

**Figure 9: Ocean surface velocities: mean zonal (top left) and mean meridional (top right) components, along with the transient zonal (middle left) and transient meridional (middle right) components over 2016. The ocean vorticity and divergence are shown in the bottom left and right panels.**

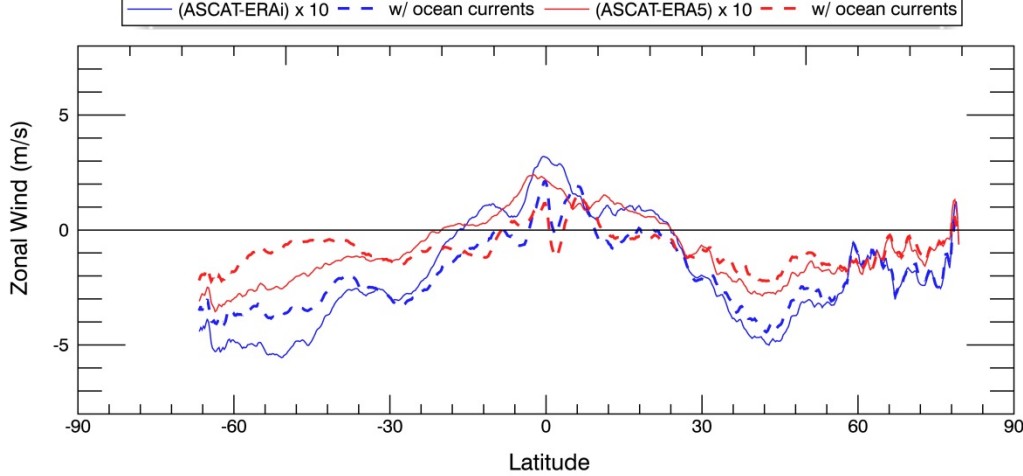

**Figure 10: Zonally averaged mean zonal wind differences between ASCAT and differences to ERA-Interim (blue line) / ERA5 (red line) first-guess winds over 2016, before (continuous lines) and after (dashed lines) the ocean current correction. Note that differences have been scaled by 10.**

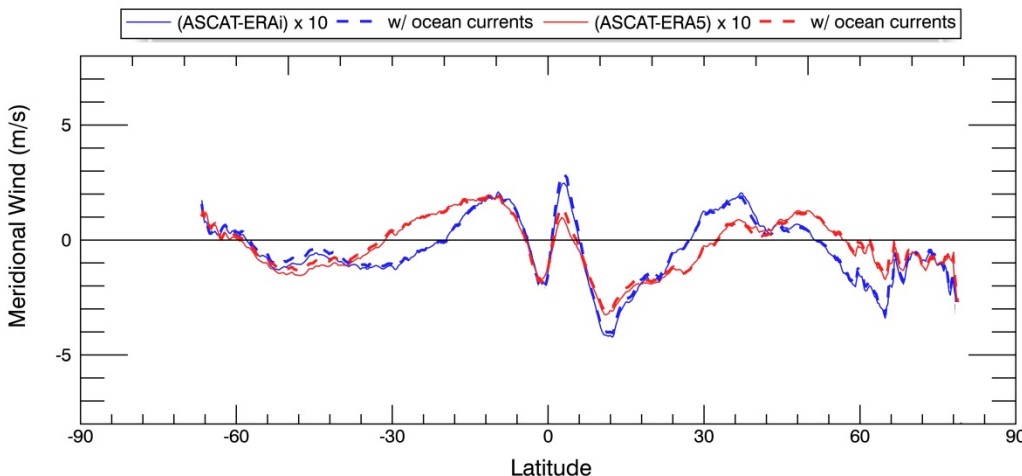

**Figure 11: Zonally averaged mean meridional wind differences between ASCAT and ERA-Interim (blue line) / ERA5 (red line) first-guess winds over 2016, before (continuous lines) and after (dashed lines) the ocean current correction. Note that differences have been scaled by 10.**

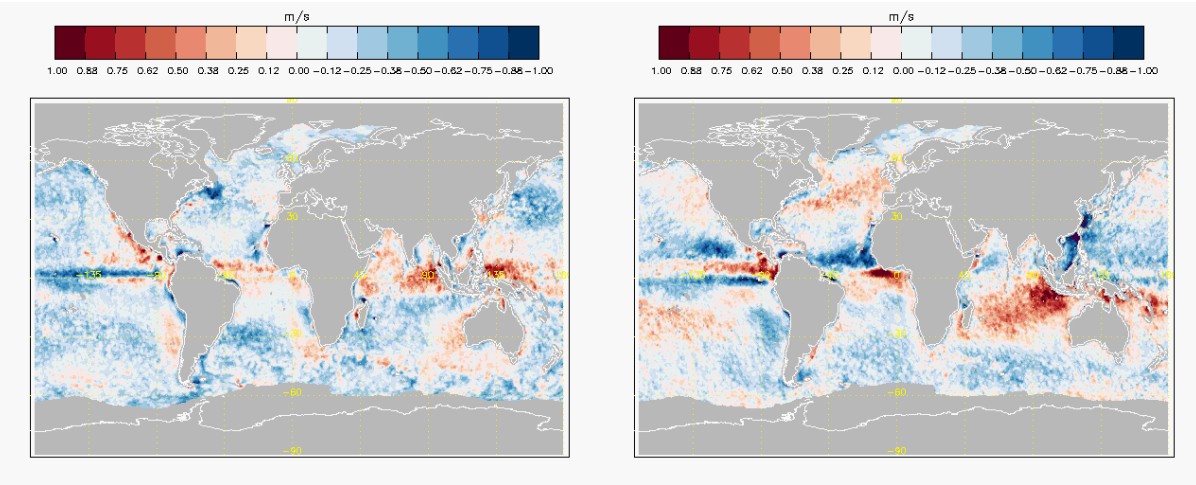

**Figure 12: Zonal (left column) and meridional (right column) mean wind differences between ASCAT and ERA5 first-guess winds over 2016 after ocean current correction.**

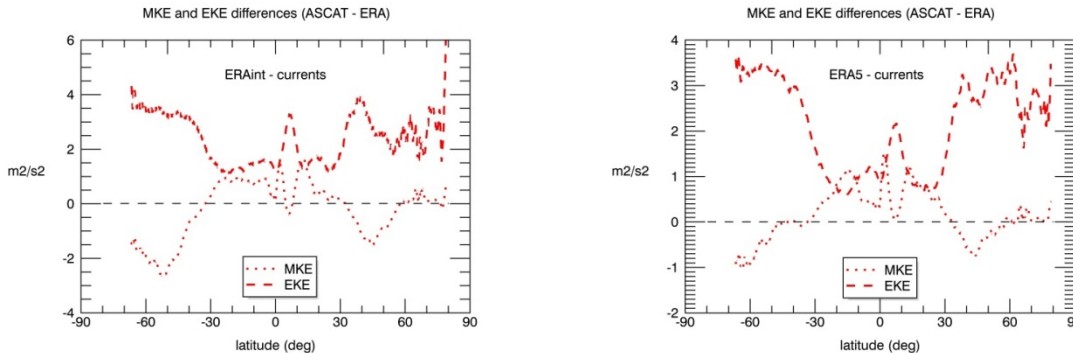

Figure 13: Zonally averaged differences in mean kinetic energy (MKE, dots) and eddy kinetic energy (EKE, dashes) between ASCAT observations and ERA-Interim (left plot) / ERA5 (right plot) first-guess winds over 2016 after the ocean current correction.

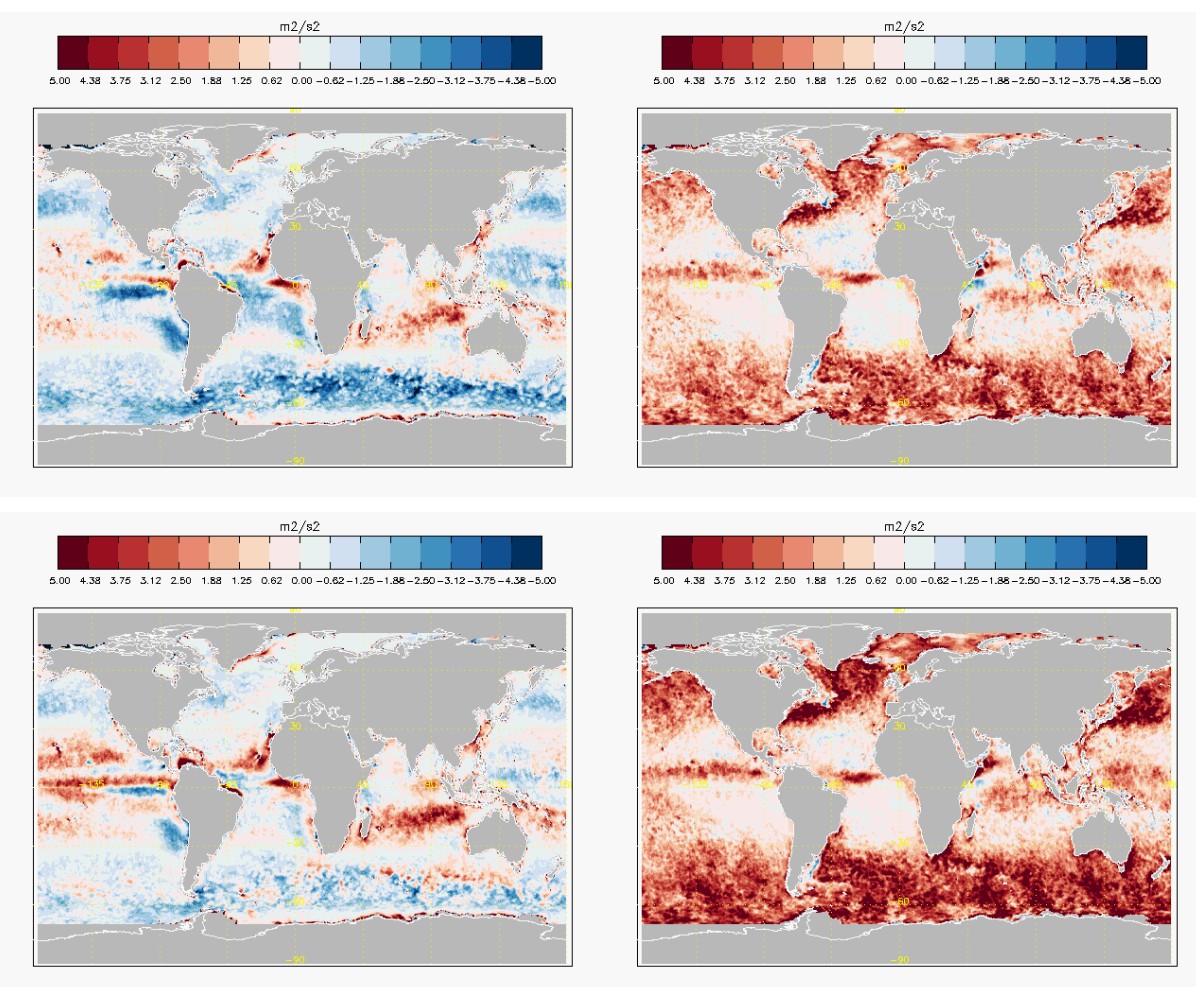

**Figure 14: MKE (left column) and EKE (right column) differences between ASCAT and ERA5 before (top row) and after (bottom row) ocean current correction over 2016.**

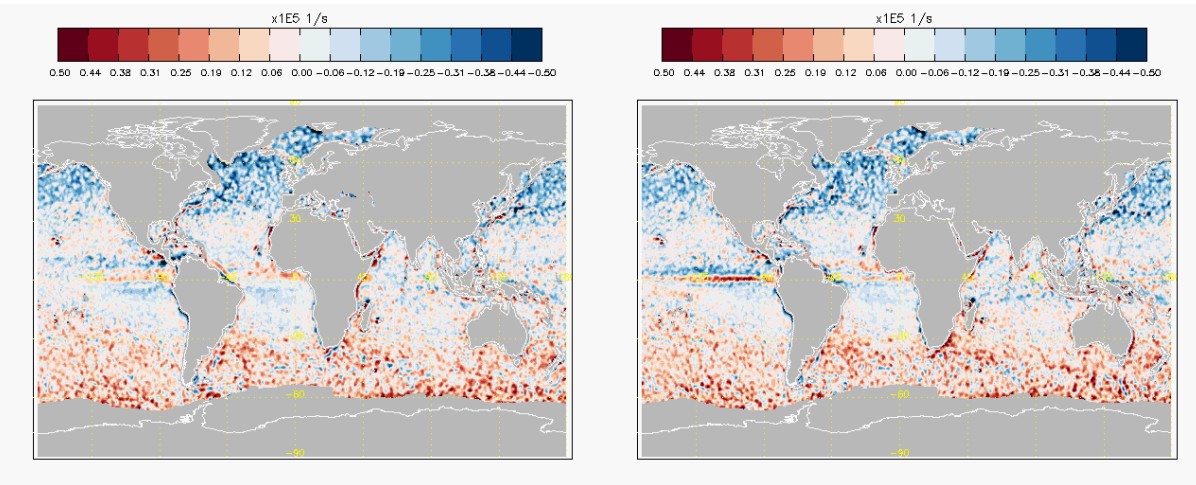

**Figure 15: Global maps of annual difference in mean wind curl between ASCAT and ERA5 before (left) and after (right) the ocean current correction.**

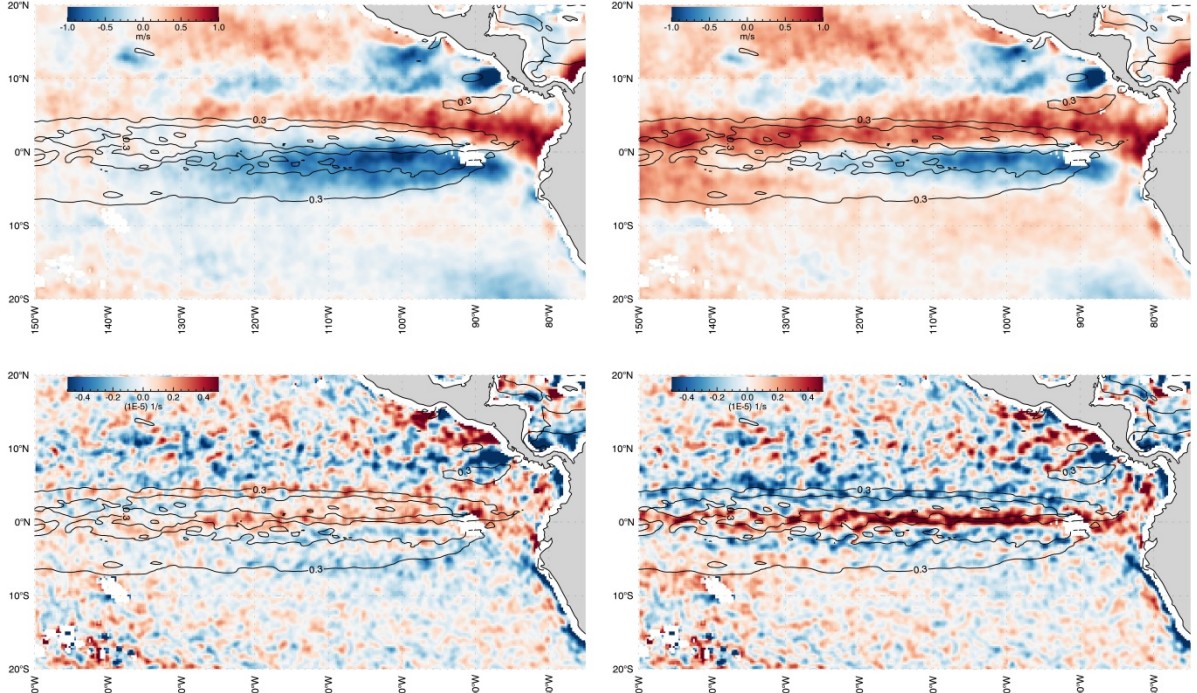

**Figure 16: Eastern tropical Pacific: annual differences in mean wind speed (top) and mean stress curl (bottom) between ASCAT and ERA5 before (left) and after (right) the ocean current correction. The contours are annual mean ocean surface velocities.**

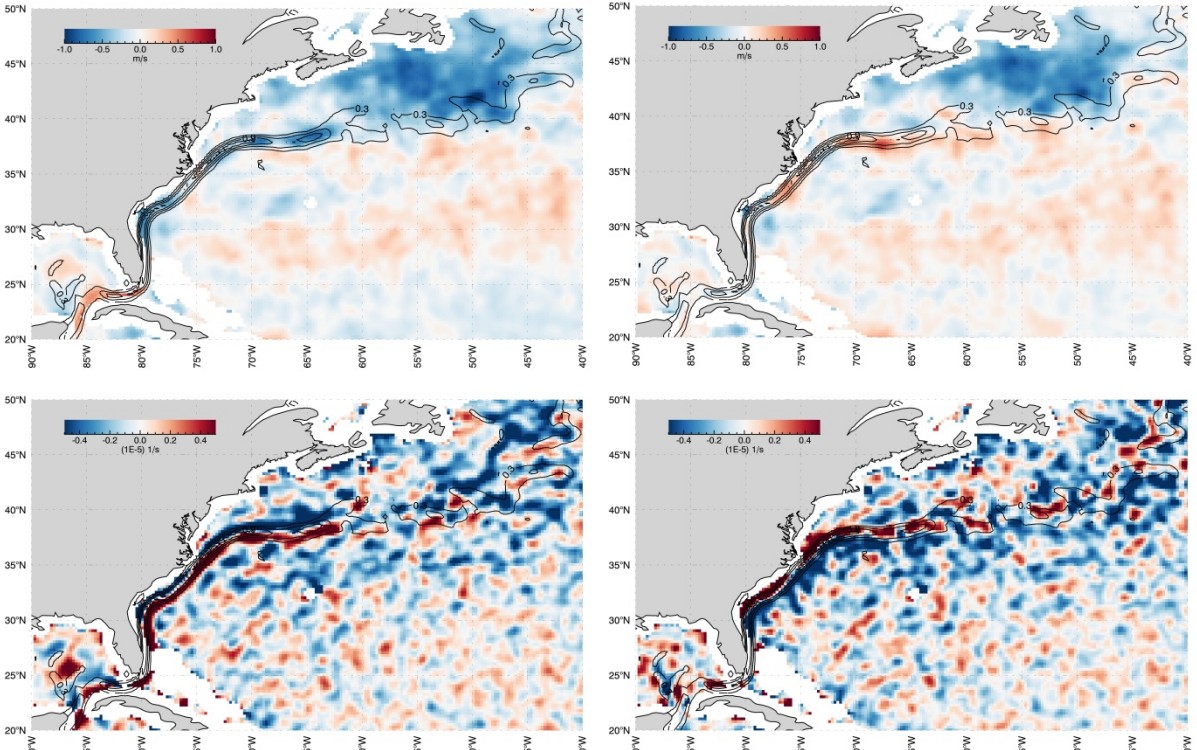

**Figure 17: Gulf stream: annual differences in mean wind speed (top) and mean stress curl (bottom) between ASCAT and ERA5 before (left) and after (right) the ocean current correction. The contours are annual mean ocean surface velocities.**

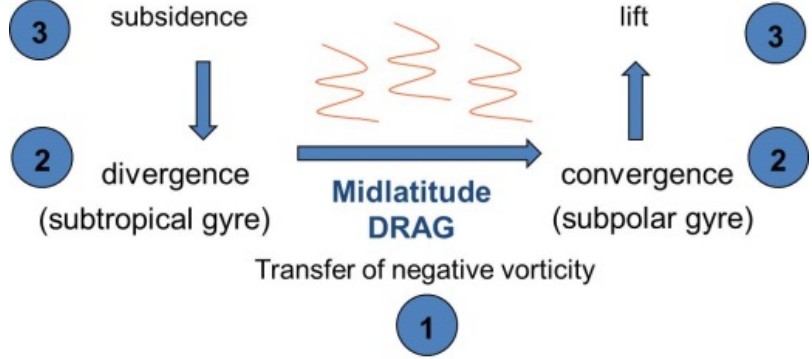

**Figure 18: Schematics of proposed residual circulation at mid-latitudes: additional mid-latitude drag would correct for (1) the zonal, meridional and stress curl errors, (2) the wind divergence errors and (3) close the circulation by vertical motions.**