# Peer review of "Characterizing ERA-interim and ERA5 surface wind biases using ASCAT"

_Ocean Science, 2018_

## Referee Comment (RC1) · Anonymous Referee #1 · 16 Apr 2019

he investigation of the quality and accuracy of numerical model (NWP) wind estimates is highly required by the oceanic and atmospheric community. ERA Interim is one the most used model for forcing and air-sea interaction process. ERA-5 is a new ECMWF re-analysis assumed improving ERA Interim data, including surface parameter estimates. In this paper, the authors investigate the comparison of both model winds, and the related variables, to ASCAT L3 product. Although, several papers and studies (some are referenced in this paper) came out with results aiming at the characterization of difference between NWP (including ERA Interim) and scatterometer wind products (L2 trough L4) at various space and temporal scales, this study is relevant and useful for the scientific community, especially interested in using ERA5 data. Throughout the paper, the authors use ASCAT L3 data as reference over global oceans. Such assumption should be better addressed in the revision version. The latter would include the impact of ASCAT L3 sampling on the comparison of wind and of the derivative data (divergence and curl). Even though the determination of wind divergence and stress curl accuracy is quite hard task, it would be interesting to assess the patterns of theses parameters through comparisons with those obtained from independent remotely sources (e.g. QuikSCAT along common period (2007 – 2009)). I think the work deserves its publication after minor revisions. $\hat{A}$

**Specific comments**

aĂć Page 2; Equation (1): I would say that few papers dealing with the calibration and validation of surface winds do use such kind of metric. It is quite common to estimate separately zonal and meridional RMS metrics. aAć Page 3; 2nd paragraph (Lines 6 -8): One may understand that upwelling dynamic relies mainly on the annual (steady) wind component. This is completely wrong. aAc Page 3; Eq (6): Obviously RMS1 is different of RMS2 in sense that the former only deals with one component (zonal in this case), while the latter aims at the characterization of difference based on the use of the two wind components. aĂć Page 5; Lines 23 - 25: The calculation of the inter-annual variability from 2016 collocated data is not clear. ć Page 7; Last paragraph: did the author investigate the impact of ASCAT sampling on eddy patterns shown in Figure 6. aĂć Page 7; Lines 21 - 23: The figure 7 should be improved. The result mentioned by the authors and dealing with the underrepresentation of the equatorial stress curl is not clear, at least the difference is very small and somehow not significant. Please clarify. âAć Page 8; Figure 9 labelling requires improvement. Thanks. âAć Page 9; Lines7 – 8: The CMEMS correction for ocean surface velocity requires details. For instance, the authors should clarify the spatial and temporal scales of correction. aÅć Page 9; Figure 12 exhibits interesting pattern along equator. Positive and negative zonal difference are found along the Atlantic and pacific equator zones, respectively. These pattern enhance the difference shown in Figure 5. Any explanations?

СЗ

---

## Referee Comment (RC2) · Mark Bourassa (Referee) · 25 Apr 2019

Summary: The paper makes great use of a time honored type of analysis to examine wind in satellites and models. The results are scientifically interesting as well as illuminating strengths and weakness of the ERA-interim and ERA5. The quality and clarity of the work are largely excellent, although there are a few places where more cautious conclusions should be drawn. The link to currents is remarkable. Major Comments:

1) Abstract: the word 'defective' carries very negative impressions, and is not very descriptive. Please use a more effective word.

2) Page 2, line 20: How are the satellite and model winds collocated? What is done to make the locations and times match? This is particularly important for metrics such as

the one at this part of the ms and on page 3, line 16.

3) Given that satellite winds tend to sample twice a day, the sampling of the diurnal cycle is regionally biases to certain times of day, which might not be representative of the other times of day.

4) Page 6, line 14: PBL stability is one of the two popular explanations for SST-related variations in wind vectors. Theory and observations indicate that this explanation is insufficient, and at least one other mechanism must be important (O'Neill et al. 2012)

5) Page 9, discussion of Figs. 13& 14. Please also discussion EKE changes in the Southern Ocean.

6) Page 11, around line 10: A more careful hypothesis is that Monin-Obukhov parameterizations are insufficient to explain the mixing in the lower atmosphere. While they are observed to work well near the surface, additional processes might be needed at in the mid and upper boundary-layer.

Minor Comments:

1) Page 1, Line 24: separate references with a comma.

2) Figure 5 caption: Should 'differences to ERA Interim' be 'differences from ERA Interim?' The same question applies to the following figures.

3) Page 10, line 4: change portray to portrait or portrayal.

---

## Author Comment (AC2) · 20 May 2019

**Replies to Referee Comments**

**OSD manuscript number os-2018-160 "Characterizing ERA-interim and ERA5 surface wind biases using ASCAT" by Maria Belmonte Rivas and Ad Stoffelen**

We are thankful to the reviewers for the time they took to read the manuscript and the judicious comments they made, which helped to improve the manuscript. Below, comments from referees are reproduced in black, our responses in red, and the changes introduced in the manuscript in green.

**Referee #2 (Mark Bourassa)**

Summary: The paper makes great use of a time honored type of analysis to examine wind in satellites and models. The results are scientifically interesting as well as illuminating strengths and weakness of the ERA-interim and ERA5. The quality and clarity of the work are largely excellent, although there are a few places where more cautious conclusions should be drawn. The link to currents is remarkable.

Major Comments:

1) Abstract: the word 'defective' carries very negative impressions, and is not very descriptive. Please use a more effective word.

OK. We have replaced the word 'defective' by 'insufficient' or "too weak".

2) Page 2, line 20: How are the satellite and model winds collocated? What is done to make the locations and times match? This is particularly important for metrics such as the one at this part of the ms and on page 3, line 16.

Agreed. The original sentence in page 4 last paragraph (section 2.2 ERA surface winds):

"The ERA-Interim first-guess winds, featuring a spatial grid of 79 km, come from 3-hourly forecasts based on 12-hourly analyses centered at 0 and 12 UTC. The ERA5 first guess winds come from 1-hourly forecasts based on 12-hourly analyses centered at 6 and 18 UTC, with an improved spatial grid of 31 km"

Is extended with:

"The model wind vector components are quadratically interpolated in time and linearly interpolated in space to match the ASCAT satellite observations."

3) Given that satellite winds tend to sample twice a day, the sampling of the diurnal cycle is regionally biased to certain times of day, which might not be representative of the other times of day.

Agreed. This issue of limited representativity of ascending measurements is very relevant, and it is now introduced in the manuscript (section 2.1, see reviewer 1 response, and last paragraph in Section 4 Discussion).

"In all cases, we note that the model-to-satellite wind differences are limited to ascending ASCAT measurements, which correspond to nighttime (approximately 9.30 pm) conditions. The diurnal variability of surface winds certainly limits the representativity of the nighttime differences that we observe, since the ERA diurnal cycle may not be perfect. Nevertheless, the main conclusions are not expected to change for daytime conditions. Actually, the boundary layer destabilization that generally takes place during daytime is expected to increase the amount of higher frequency wind variability, which is generally underestimated by ERA, and thus enhance the magnitude of the model-to-satellite differences reported here."

4) Page 6, line 14: PBL stability is one of the two popular explanations for SST-related variations in wind vectors. Theory and observations indicate that this explanation is insufficient, and at least one other mechanism must be important (O'Neill et al. 2012)

Agreed. The original sentence:

"The SST-gradient effect describes how surface winds vary in response to SST modification of atmospheric boundary layer stability via PBL destabilization with heat fluxes."

Is now replaced by:

"The SST-gradient effect describes how surface winds dynamically respond to SST modification and associated ocean heat flux changes (O'Neill, 2012; Skyllingstad et al., 2007)."

O'Neill, L. W., 2012: Wind speed and stability effects on coupling between surface wind stress and SST observed from buoys and satellite, J. Climate, 25, 1544-1569, doi: 10.1175/JCLI-D-11-00121.1

Skyllingstad, Eric D., Vickers, Dean, Mahrt, Larry, Samelson, Roger, 2007, Effects of mesoscale sea-surface temperature fronts on the marine atmospheric boundary layer, Boundary-Layer Meteorology 123 (2), 219 – 237, https://doi.org/10.1007/s10546-006-9127-8

5) Page 9, discussion of Figs. 13& 14. Please also discussion EKE changes in the Southern Ocean.

The original paragraph:

"Figures 13-14 show the balance between observed and model mean and eddy kinetic energies before and after the ocean current correction (cf. Fig.2). The alleviation of zonal

mean wind errors reduces the MKE differences in the mid-latitudes by about one half, but increases the MKE differences in the tropics, indicating that the model mean wind speeds in the trade regions have become weak relative to observations after the ocean current correction, which is to be mainly attributed to defective model meridional inflows into the ITCZ (see right panel in Fig.12). We observe that EKE differences increase globally, particularly in the extra-tropics. "

Has been extended with:

"In the Southern Ocean, the increase in EKE differences is accompanied by the largest decrease in MKE differences."

6) Page 11, around line 10: A more careful hypothesis is that Monin-Obukhov parameterizations are insufficient to explain the mixing in the lower atmosphere. While they are observed to work well near the surface, additional processes might be needed at in the mid and upper boundary-layer.

**Agreed. The original paragraph has been extended with:**

"The Monin-Obukhov parameterizations are observed to work well near the surface, but additional processes might be needed at higher levels in the boundary layer."

**Minor Comments:**

1) Page 1, Line 24: separate references with a comma.

**Done**

2) Figure 5 caption: Should 'differences to ERA Interim' be 'differences from ERA Interim?' The same question applies to the following figures.

**Done**

3) Page 10, line 4: change portray to portrait or portrayal.

**Done**

---

## Author Comment (AC1)

**Replies to Referee Comments**

**OSD manuscript number os-2018-160 "Characterizing ERA-interim and ERA5 surface wind biases using ASCAT" by Maria Belmonte Rivas and Ad Stoffelen**

We are thankful to the reviewers for the time they took to read the manuscript and the judicious comments they made, which helped to improve the manuscript. Below, comments from referees are reproduced in black, our responses in red, and the changes introduced in the manuscript in green.

**Anonymous Referee #1**

The investigation of the quality and accuracy of numerical model (NWP) wind estimates is highly required by the oceanic and atmospheric community. ERA Interim is one the most used model for forcing and air-sea interaction process. ERA-5 is a new ECMWF re-analysis assumed improving ERA Interim data, including surface parameter estimates.

In this paper, the authors investigate the comparison of both model winds, and the related variables, to ASCAT L3 product. Although, several papers and studies (some are referenced in this paper) came out with results aiming at the characterization of difference between NWP (including ERA Interim) and scatterometer wind products (L2 trough L4) at various space and temporal scales, this study is relevant and useful for the scientific community, especially interested in using ERA5 data.

Major comments:

1) Throughout the paper, the authors use ASCAT L3 data as reference over global oceans. Such assumption should be better addressed in the revision version.

That is correct. The ASCAT L3 data is used as a reference over the global oceans. We added to the ASCAT paragraph:

Note that ascending orbits correspond to a local solar time equator crossing (LTAN) of the sun-synchronous MetOp satellite of 21:30 in the evening. This is, the ERA diurnal cycle is only collocated and differenced around this time of day, though without any time or space sampling errors.

2) The revision version would include the impact of ASCAT L3 sampling on the comparison of wind and of the derivative data (divergence and curl).

The impact of satellite sampling is negligible – because model and satellite winds are space and time collocated in order to remove sampling errors. This is now further emphasized by the added text quoted here above. The remaining geophysical differences do include differences in smoothness of ERA and ASCAT, since this is, inter alia, expressed in the spatial derivative data (divergence, curl). We added: We further note that the effective spatial resolution of ASCAT is about 25 km, while that of the best model product is less than 100 km [Vogelzang et al., 2011], which obviously will impact spatial gradient amplitudes.

Please, see also our reply to Reviewer 2 comment: "did the author investigate the impact of ASCAT sampling on eddy patterns?"

3) Even though the determination of wind divergence and stress curl accuracy is quite hard task, it would be interesting to assess the patterns of theses parameters through comparisons with those obtained from independent remotely sources (e.g. QuikSCAT along common period (2007 – 2009)).

Sure, an interesting exercise like this should belong in a paper that analyzes satellite sampling effects on wind field derivatives. In our case, satellite sampling effects have been removed from the equation using space and time collocation of satellite and model winds (which in our case always takes place before the field derivatives are estimated).

Exploiting collocated Ku and C band winds, other authors have elaborated on SST effects in Ku-band scattering. Nonetheless, time-average collocated differences of ASCAT-EC and RapidScat-EC look very similar indeed and are about twice larger in amplitude than ASCAT-RapidScat differences, i.e., including the SST artifacts. See file:

https://www.eumetsat.int/website/wcm/idc/idcplg?IdcService=GET\_FILE&dDocName=P DF\_TL\_15\_09\_16\_B&RevisionSelectionMethod=LatestReleased&Rendition=Web

Specific comments

Page 2; Equation (1): I would say that few papers dealing with the calibration and validation of surface winds do use such kind of metric. It is quite common to estimate separately zonal and meridional RMS metrics.

Ok. We have softened the statement. The sentence:

"One of the statistical metrics most widely used to assess..."

Is now replaced by:

"A common statistical metric used to assess..."

Page 3; 2nd paragraph (Lines 6-8): One may understand that upwelling dynamic relies mainly on the annual (steady) wind component. This is completely wrong.

Transient wind stresses lasting more than a day have been observed to lead to transient upwelling or downwelling events. In order to avoid confusion, the original sentence:

"One may consider how the total wind energy at a given location is partitioned into separate mean (steady) and transient (eddy) components, since they affect the ocean circulation and its gyres differently, the former through large-scale Ekman transport and upwelling, the latter through vertical mixing via wave motion, inertial currents, etc."

Is now replaced by:

"One may consider how the total wind energy at a given location is partitioned into separate mean (steady) and transient (eddy) components, since they affect the ocean circulation and its gyres differently. Steady wind stresses are associated with large-scale upwelling/downwelling and Ekman transport in the global oceans. Transient wind stresses, which are associated with the development of surface and internal wave motions, inertial currents and transient upwelling/downwelling events, mainly contribute in a timeintegral sense to vertical mixing and the development of the mixed layer."

Page 3; Eq (6): Obviously RMS1 is different of RMS2 in sense that the former only deals with one component (zonal in this case), while the latter aims at the characterization of difference based on the use of the two wind components.

Thanks for noticing. Equation (6) was missing the meridional terms. The original Equation (6):

$$RMS_{2} = \sqrt{(u_{m,scat} - u_{m,nwp})^{2} + (u_{e,scat} - u_{e,nwp})^{2}}$$

Is now replaced by:

 $RMS_{2} = \sqrt{\left(u_{m,scat} - u_{m,nwp}\right)^{2} + \left(u_{e,scat} - u_{e,nwp}\right)^{2} + \left(v_{m,scat} - v_{m,nwp}\right)^{2} + \left(v_{e,scat} - v_{e,nwp}\right)^{2}}$

Page 5; Lines 23 – 25: The calculation of the inter-annual variability from 2016 collocated data is not clear.

The inter-annual variability of zonally averaged mean (ASCAT-ERAint) wind differences is not analyzed or shown in the manuscript, but it can be verified here:

https://mdc.coaps.fsu.edu/scatterometry/meeting/docs/2017/docs/Posters/OceanState. pdf

In any case, the original sentence:

"The systematic mean differences are very stable in time, with an inter-annual variability of about 0.1 m/s ..."

Is replaced by:

"The systematic mean differences are very stable in time, with an inter-annual variability of about 0.1 m/s (not shown) ..."

Page 6; Last paragraph: did the author investigate the impact of ASCAT sampling on eddy patterns shown in Figure 6.

We added a new paragraph as we indeed agree that spatial sampling and effective resolution are prime parameters for readers to understand:

**3.1 Effect of spatial resolution**

The amount of energy captured in the eddy maps of Fig. 6 obviously depends on spatial resolution, which for the ASCAT data is approximately 25 km. Recall that the ERA effective spatial resolution is less than 100 km [Vogelzang et al., 2011].

All the ERA products have been carefully collocated in space and time to match ASCAT observations, but the differences in effective spatial resolution remain problematic, not so much for the mean wind or mean derivative fields, but for the eddy fields.

To further investigate this effect a spatial smoothing, using a 1.5 degree spatial filter before calculating the mean and eddy quantities, was employed. It appeared (not shown) that indeed mean (wind, curl, divergence) differences are not appreciably dependent on resolution, including transient (eddy) wind differences. On the other hand and as anticipated, transient (eddy) curl and divergence are affected by the spatial filter: both model and satellite eddy fields change, but the reduction is largest in the satellite fields. In summary, if we force the spatial resolution of both model and satellite data to be similar (down to 100-200 km), the differences in mean wind/curl/divergence do not change. That is, meridional winds are still weak in the model, tropical convergence is still low in the model and stress curl at high latitudes is still high in the model.

The key to understanding these differences seems to lie in the representation of the higher frequency transient eddy winds. The application of a spatial filter only obscures that conclusion.

Page 7; Lines 21 – 23: The figure 7 should be improved. The result mentioned by the authors and dealing with the underrepresentation of the equatorial stress curl is not clear, at least the difference is very small and somehow not significant. Please clarify.

We agree that the underrepresentation of model stress curl over the equatorial cold tongues is not as large or striking as that observed at higher latitudes, yet it drew our attention. A more detailed (zoomed) depiction of this effect in the eastern tropical Pacific is shown later in the manuscript in Figure 16. The original sentence:

"The map of mean stress curl differences indicate that the signature of wind curl associated with the equatorial cold tongue is underrepresented in the ERA products, with model defective positive curl in the northern front (reddish color) and model defective negative curl in the southern front (blueish color, see bottom left panels in Fig.7)."

Has been extended to include a more explicit reference to Figure 16.

"A more detailed (zoomed) depiction of the underrepresentation of model stress curl over the cold tongue in the eastern tropical Pacific is deferred to Figure 16."

Page 8; Figure 9 labelling requires improvement. Thanks.

The Figure 9 caption has been corrected. Thanks.

"Figure 9: Ocean surface velocities: mean zonal (top left) and mean meridional (top right) components, along with the transient zonal (middle left) and transient meridional (middle right) components over 2016. The ocean vorticity and divergence are shown in the bottom left and right panels."

Page 9; Lines7 – 8: The CMEMS correction for ocean surface velocity requires details. For instance, the authors should clarify the spatial and temporal scales of correction.

Sure. The original sentence:

"Figures 10 and 11 show the zonal and meridional mean wind differences between the reanalysis and scatterometer products obtained after applying the CMEMS correction for ocean surface velocity."

Has been extended with:

"The CMEMS ocean current correction is derived from the original 3 hourly fields of zonal and meridional ocean surface velocities, gridded at 25 km, which are linearly interpolated in space and time to match ASCAT observations."

Page 9; Figure 12 exhibits interesting pattern along equator. Positive and negative zonal differences are found along the Atlantic and pacific equator zones, respectively. These patterns enhance the difference shown in Figure 5. Any explanations?

The positive and negative zonal wind differences along the Atlantic and Pacific equator zones seen in Figure 5 and Figure 12 are associated with SST-gradient effects over the equatorial cold tongues. The SST-gradient effect describes how surface winds, which vary in response to ocean heat fluxes, decelerate/accelerate as they straddle into cold/warm waters.

The ocean current correction is introducing quite an amount of new structure in the wind differences observed over equatorial cold tongues, enhancing the SST-gradient related

differences in a manner such that the wind and wind derivative differences align more realistically with the underlying SST fronts that sustain the ocean currents. Figure 16 shows a detailed (zoomed) depiction of this effect.